# The Lymph-Sparing Quotient: A Retrospective Risk Analysis on Extremity Radiation for Soft Tissue Sarcoma Treatment

**DOI:** 10.3390/cancers13092113

**Published:** 2021-04-27

**Authors:** Iqbal Sarif, Khaled Elsayad, Daniel Rolf, Christopher Kittel, Georg Gosheger, Eva Wardelmann, Uwe Haverkamp, Hans Theodor Eich

**Affiliations:** 1Department of Radiation Oncology, University Hospital Muenster, 48149 Munster, Germany; khaled.elsayad@ukmuenster.de (K.E.); daniel.rolf@ukmuenster.de (D.R.); Christopher.Kittel@ukmuenster.de (C.K.); Uwe.Haverkamp@ukmuenster.de (U.H.); hans.eich@ukmuenster.de (H.T.E.); 2Department of Orthopedics and Tumor Orthopedics, University Hospital Muenster, 48149 Muenster, Germany; georg.gosheger@ukmuenster.de; 3Gerhard Domagk Institute of Pathology, University Hospital Muenster, 48149 Muenster, Germany; eva.wardelmann@ukmuenster.de

**Keywords:** radiotherapy, soft tissue sarcoma, lymphedema, toxicity, personalized treatment, risk-based treatment

## Abstract

**Simple Summary:**

Soft tissue sarcomas, a heterogenous group of tumors with a mesenchymal origin, are mostly located in the extremities and are commonly treated with surgery and radiotherapy. Using opportunities of reducing long-term therapy-related side effects in soft tissue sarcoma treatment is an important task for all physicians involved in soft tissue sarcoma treatment. The extent of lymph-sparing volume in adjuvant radiation therapy of extremity soft tissue sarcoma as a risk factor for lymphedema was analyzed in this study. Patients with a low lymph-sparing volume showed an increased risk of lymphedema in this retrospective study. Maximizing the potential oncologically justifiable lymph-sparing volume should be considered to reduce the risk of high-grade lymphedema when applying RT to extremities.

**Abstract:**

Radiation therapy (RT) for extremity soft tissue sarcoma is associated with lymphedema risk. In this study, we analyzed the influence of lymph-sparing volume on the lymphedema occurrence in patients who received adjuvant extremity RT. The lymph-sparing quotient (LSQ) was calculated by dividing the lymph-sparing volume by the total extremity volume with double weightingfor the narrowest lymph-sparing region. A total of 34 patients were enrolled in this analysis. The median applied total radiation dose was 66.3 Gy in 36 fractions. Acute lymphedema appeared in 12 patients (35%). Most of them (*n* = 8) were lymphedema grade 1 and five patients had grade 2 to 3 lymphedema. Chronic lymphedema appeared in 22 patients (65%). 17 of these patients had at least a grade 2 lymphedema. In 13 of 14 patients with an LSQ ≤ 0.2 and 11 of 20 patients with an LSQ > 0.2, an acute or chronic lymphedema ≥ grade 2 was observed. A Kaplan–Meier Analysis of the two groups with the endpoint of a two-year lymph edema-free survival (=2-YLEFS) was estimated with an univariate, significant result (2-YLEFS LSQ ≤ 0.2 vs. LSQ > 0.2: 0% vs. 39%; *p* = 0.006; hazard ratio LSQ ≤ 0.2 vs. > 0.2 2-YLEFS 2.822 (*p* = 0.013); 95% confidence interval (CI): 1.24–6.42). Maximizing the potential oncologically-justifiable lymph-sparing volume should be considered to reduce the risk of high-grade lymphedema when applying RT to extremities.

## 1. Introduction

Soft tissue sarcomas, a heterogenous group of tumors with a mesenchymal origin, are mostly located in the extremities and are commonly treated with surgery and radiotherapy [1].

The use of radiotherapy in soft tissue sarcoma patients improves local control and overall survival [2,3]. The complete resection of the tumor with negative margins is important for the mutual success of surgery and radiotherapy [4,5]. However, marginal and intralesionally-resected sarcoma patients have shown better local control when post-operatively irradiated [4,5].

Different prospective randomized studies have underlined the benefit of radiotherapy in terms of local control in soft tissue sarcoma patients [6,7].

The use of adjuvant chemotherapy in soft tissue sarcoma patients remains controversial. However, a combination of Doxorubicin and Ifosfamid in high-risk patients can be an option and their use depends on various circumstances [8]. Accordingly, treatment must be individualized and discussed in the interdisciplinary tumor board.

Postoperative RT is associated with lymphedema [9], which can possibly lead to long-term treatment necessity and a negative influence on the quality of life [10].

The lymphatic system is essential for the circulation of immune cells and protein and fluid (filtrated blood) from the interstitium to the blood circulation. Lymphedema is a chronically inflammatory lymphatic disease caused by the mechanical insufficiency of the lymph vessel system [11]. When lymphatic drainage is impeded (by, for example, radiation or an operation) a regional accumulation of lipids and proteins can lead to local fibrosis, fat deposition and inflammation [12].

In this study, we analyzed the influence of lymph-sparing volume on the occurrence of lymphedema in patients who received adjuvant extremity RT as a part of extremity soft tissue sarcoma treatment.

## 2. Methods

In this study, 34 patients with extremity soft tissue sarcoma treated with adjuvant RT in our department for Radiation Oncology of the University Hospital of Muenster from 2000 to 2014 were retrospectively analyzed. The patients were adults and had not had prior RT. Radiation was part of a multidisciplinary treatment and a continuous lymph-sparing volume could be seen in the radiation plan. The endpoint of this univariate analysis was an acute or chronic lymphedema ≥ grade 2 up to two years after radiation. The two-year lymphedema-free survival (2-YLEFS) corresponded with the probability of not having an acute or chronic lymphedema ≥ grade 2 up to two years after radiation. As the 40 month median follow up represented a range from 3 to 223 months we decided to limit the endpoint to a two year period after therapy to ensure most patients reached this follow up period.

### The Lymph-Sparing Quotient

The irradiated part of the extremity was demonstrated in the radiation plan including the isodose lines to confirm a continuous lymph-sparing volume outside the 20% isodose level. If a boost was applied, the plan sum was considered. Depending on the extent of the radiation field, the extremity was defined from the middle of the ankle joint up to the hip joint or from the radioulnar joint up to the shoulder joint, respectively. The extremity and the continuous lymph-sparing volume were created as a 3D structure in the planning CT. In a few parts of the extremity the lymph-sparing volume was narrower due to anatomical and disease-specific reasons (Figure 1 and Figure 2). Bone volume was subtracted from the extremity volume (Figure 3 and Figure 4). The narrowest 10 cm of the lymph-sparing volume was also defined as a 3D structure (Figure 5).

The next step was to calculate the proportion of the lymph-sparing volume in relation to the extremity volume. The same was done with the narrowest part of the lymph-sparing volume, which was double weighted in the formula for the lymph-sparing quotient (Figure 6).
$$\frac{1}{3}\times \frac{LSV\;total}{V\;extremity}+\frac{2}{3}\times \frac{\left(LSV\;narrowest\;10\;\mathrm{cm}\right)}{V\;ex.\;narrowest\;10\;\mathrm{cm}}=LSQ$$

LSV = lymph-sparing volume, V = volume and LSQ = lymph-sparing quotient.

Depending on the calculated median of 0.23, the patients were divided into two groups: lymph-sparing quotient ≤ 0.2 and > 0.2. With the documented follow up examinations and a Kaplan–Meier analysis/log-rank test/Cox regression, the outcomes of the two groups regarding 2-YLEFS were analyzed. All statistical analyses were conducted with IBM SPSS Statistics 25.0 software (SPSS Inc., Chicago, IL, USA). Differences were considered statistically significant at *p* < 0.05.

The graduation of observed lymphedema depended on the extent, the necessity of therapy and symptoms with the following scale: scale 0: no lymphedema, scale 1: low grade lymphedema with only mechanical stress and no symptoms, scale 2: moderate lymphedema with all day mild symptoms, scale 3: high-grade lymphedema with all day pain and therapy necessary, scale 4: massive lymphedema with exudation and severe pain.

## 3. Results

In 24 of the 34 patients, an acute or chronic lymphedema ≥ grade 2 occurred up to two years after radiation (71%). Of the 34 patients, 32 (94%) had a lower extremity sarcoma and 2 (6%) had an upper extremity sarcoma. Acute lymphedema (within three months after radiation) was seen in 12 patients. Most of them were grade 1 lymphedema (7 = 58%) while five patients (42%) had a grade 2 or 3 acute lymphedema. Chronic lymphedema (three months after therapy) occurred in 22 patients (65%). In 17 of these patients, a lymphedema ≥ grade 2 was seen (77%). A chronic grade 1 or grade 3 lymphedema was seen in two patients (6%) and six patients (18%), respectively. No grade 4 lymphedema (acute or chronic) was seen (Table 1). No patient presented a relevant post-operative lymphedema before radiation therapy treatment. Furthermore, no lymphfistula or lymphocele were diagnosed.

### Use of the Lymph-Sparing Quotient

The LSQ was calculated as described in the Methods section in 34 patients with all necessary information available. The LSQ was used as an univariate influence factor in the first place. The median value of 0.23 (standard deviation 0.13) and the average 0.25 (standard deviation 0.13) led to the separation of patient groups in LSQ ≤ 0.2 and LSQ > 0.2.

The 34 patients received a median total dose of 66.3 in 36 fractions, each 1.8 Gy. A total of 27 patients received a boost with a median dose of 16.2 Gy. Of those, 93% (13/14) of the patients in group LSQ ≤ 0.2 had a lymphedema ≥ Grade 2 whereas this was the case in only 55% (11/20) in group LSQ > 0.2. With a Kaplan–Meier analysis, it was estimated that after 24 months, 39% of the patients in group LSQ > 0.2 remained without a lymphedema ≥ grade 2 whereas this was the case in 0% of group LSQ ≤ 0.2 (*p* = 0.006). With a Cox regression, there was a hazard ratio regarding the 2-YLEFS of group LSQ ≤ 0.2 in relation to group LSQ > 0.2 of 2.822 (*p* = 0.013; 95% CI = 1.24–6.4).

Regarding the LSQ itself, the Cox regression analysis showed that increasing the LSQ by 1% resulted in a significant univariate reduction of hazard for a lymphedema ≥ Grade 2 by a factor of 0.964 (95% CI: 0.93–1. *p* = 0.05). For example, increasing the quotient by 5% points reduced the hazard by a factor of 0.83 (95% CI: 0.69–1).

## 4. Discussion

This analysis aimed to examine the impact of the LSQ on radiation-related toxicity. The following findings emerged from this analysis: (1) with the creation of the LSQ in this analysis, a possibility for graduating the risk of lymphedema ≥ grade 2 in irradiated extremity soft tissue sarcoma patients was established. (2) The results of the groups (≤0.2 vs. >0.2: 0% vs. 39%) regarding 2-YLEFS with a *p*-value of 0.006 showed univariate significant differences. Furthermore, the importance of every percent increase of lymph-sparing volume in relation to extremity volume could be concluded by showing that a 1% increase of the LSQ resulted in a hazard ratio of 0.964 lymphedema ≥ grade 2 in two years with a confidence interval of 0.93–1.0 (*p* = 0.05).

Further development of this concept could include a prospective study by calculating the LSQ before starting the radiation. If these analysis results could be confirmed in a prospective study, the possibilities for optimizing the treatment plan without influencing the dose in the planning target volume could be considered. The restrictions of the interpretation of the results of this analysis have to be demonstrated. A limitation of our study was the relatively small number of patients included.

Interobserver variability in estimating the narrowest part of the LSV was not tested. A certain amount of subjectivity cannot be denied so testing the variation of the LSQ calculation by different users of the formula could increase the objectivity of the calculation.

Most likely because of the usage of different treatment regimens and different classification scales for diagnosis of lymphedema and subjective judging, the incidence of lymphedema in studies regarding radiation of extremity soft tissue sarcoma varies.

The results regarding the incidence of lymphedema were comparable with the incidence reported by Bell et al. In this study with 88 extremity soft tissue sarcomas, 60% of the patients (27% ≥ grade 2) experienced lymphedema while or after therapy [14]. A unicentric study by Friedmann et al. analyzed the prognostic factors for the appearance of a lymphedema after treatment of extremity soft tissue sarcomas by using the Sterns rating scale (0 = no lymphedema, 1: mild but definite swelling, 2: moderate lymphedema, 3: severe (considerable swelling), 4: very severe (skin shiny and tight/skin cracking)) in 289 patients; a lymphedema incidence of 29% was seen (6% ≥ grade 2) [9].

Friedmann et al. analyzed patients who were treated in a preoperative setting while our study included only post-operative patients. Apart from that, the used radiation dose concepts differed. Friedmann et al. stated a significant univariate difference between the development of lymphedema in the group treated with 50 Gy vs. the group with 66 Gy (*p* = 0.01). This suggests that the median radiation dose of Friedmann et al. was lower than in our study, which could be a reason for a lower lymphedema incidence [9].

Different scales for lymphedema classification were used in both studies and both scales do not include objective parameters. In a different study of Stinson et al. with 145 patients treated with extremity soft tissue sarcomas, an incidence of 19% ≥ grade 2 lymphedema was observed. An odds ratio of 5 for the lymphedema appearance in lower extremity soft tissue sarcomas vs. upper extremity sarcomas was stated [15].

Three other studies analyzing extremity preserving treatment of soft tissue sarcomas from Bedi et al. (22%), Lampert et al. (45%) and Robinson et al. (30%) stated a lymphedema incidence of 22–45% [16,17,18].

It is important to discuss the different approaches of grading lymphedema in the previously mentioned studies. When referring to the lymphedema staging table from the International Society for Lymphology we can conclude that it combines pathophysiological and clinical parameters such as fluid accumulation, fibrosis and changes in edema by limb elevation [19].

The Sterns rating scale used by Friedmann et al., as with our scale, included subjective results of clinical examination by different examiners [9].

All patients included in our study received various treatment modalities including a resection of the tumor before receiving radiotherapy. Due to an oncologic tumor resection the risk of a lymphedema could increase depending on different factors, for example, medial location or vessel reconstruction [20]. Bedi et al. used the term post therapy lymphedema to underline this fact because soft tissue sarcomas of the extremities always need multimodality treatments in a curative setting. Furthermore, Bedi et al. also used the extent of affection in the activities of daily living additionally to their subjective graduation based on clinical examination in mild, moderate and severe cases [18].

To compare the results concerning lymphedema in soft tissue sarcoma treatments comparable rating scales based on objective techniques are needed. Apart from clinical examination, different objective techniques for lymphedema detection and graduation have to be discussed.

Allam et al. presented different objective techniques that were partly established in detecting and grading upper limb lymphedema after breast conserving therapy and these may be helpful for soft tissue sarcoma patients as well [21]. Tonometry, bioimpedance and the use of a perometer are different promising apparative methods to objectively diagnose lymphedema with different reliability [21]. In general their usage and potential benefit of early detection and treatment should be considered in a prospective trial with soft tissue sarcoma patients. In a study of Moseley and Piller the reproducibility of tonometry and bioimpedance measurements was positively tested [22]. Tonometry also has been used in a study to estimate the involvement of adipose tissue in therapy-related lymphedema as a factor of the potential improvement of an edema through liposuction [23].

Cornish et al. used a multiple frequency bioelectrical impedance analysis as a reliable method for the early detection of upper limb lymphedema following breast cancer treatment [24].

Furthermore, the imaging method lymphoscintigraphy (injecting Technetium 99m antimony trisulfide colloid subcutaneously) can detect abnormalities in lymphatic drainage and can make it possible to score lymphatic transport time and lymphatic vessel and node distribution [25].

In the studies of Lampert and Stinson, patients with lower extremity sarcomas showed a significantly higher probability of lymphedema while in our study there was no significant difference in lymphedema incidence between upper and lower extremity patients.

As only two patients with upper extremity sarcomas were included in our study, our results cannot be used as a reliable factor when discussing a higher risk of lymphedema in different extremity locations after multimodality treatment of soft tissue sarcomas [15,16].

When considering an additional risk for lymphedema with multimodality treatment, different options of treating and preventing lymphedema apart from the presented study have to be discussed.

An established therapeutic approach to treat lymphedema is decongestive therapy that consists of the four key aspects of skin care, lymphatic drainage, compression therapy and physiotherapy [11].

Different studies have already shown a positive influence on edema reduction of this therapeutic approach when consequently applied in lymphedema patients [26,27].

Surgical options for preventing lower limb lymphedema have been used in patients that underwent an inguinal lymphonodectomy. Lymphatic-venous anastomoses or a lymph node transfer are surgical options to prevent or treat post-therapeutic lymphedema; however, it is not a standard procedure [28,29,30].

Finally, prevention and therapy methods for lymphedema aim to reduce the negative impact on life quality and risk for rare but aggressive complications such as the Stewart–Treves syndrome, lymphangiosarcoma [31].

Our study illustrated an increased risk of lymphedema with decreasing lymph-sparing volume. The results and the application of the illustrated method should be addressed by a multi-institutional prospective study.

In the era of targeted therapies, the additional investigation in terms of radiogenomic and personalized medicine is needed to optimize therapy choices.

Different immunotherapy approaches in high-risk sarcoma patients include, for example, the promising use of the checkpoint inhibitor pembrolizumab in undifferentiated pleomorphic sarcomas or liposarcomas and further adaptive cell transfer concepts, for example, the intratumoral injection of dendritic cells [32,33].

## 5. Conclusions

The effective treatment of localized extremity soft tissue sarcomas consists of a wide resection and radiation with sufficient margins. Lymphedema is a common side effect after a combined modality treatment. Maximizing the potential oncologically-justifiable lymph-sparing volume should be considered to reduce the risk of high-grade lymphedema when giving radiation to extremities in curative soft tissue sarcoma treatment settings.

## Figures and Tables

**Figure 1 cancers-13-02113-f001:**
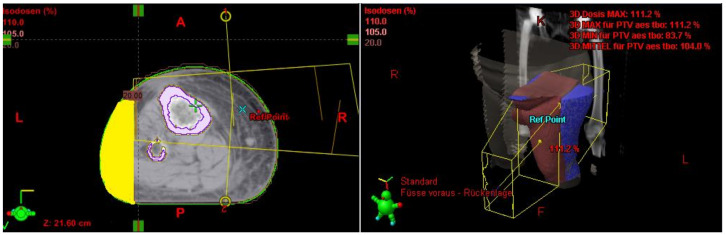
Relatively wide part of the lymph-sparing volume (yellow) outside the 20% isodose line.

**Figure 2 cancers-13-02113-f002:**
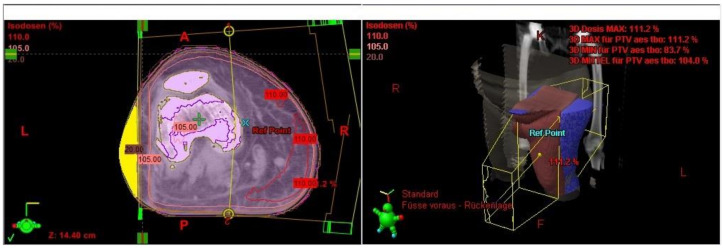
Relatively narrow part of the lymph-sparing volume (yellow) outside the 20% isodose line.

**Figure 3 cancers-13-02113-f003:**
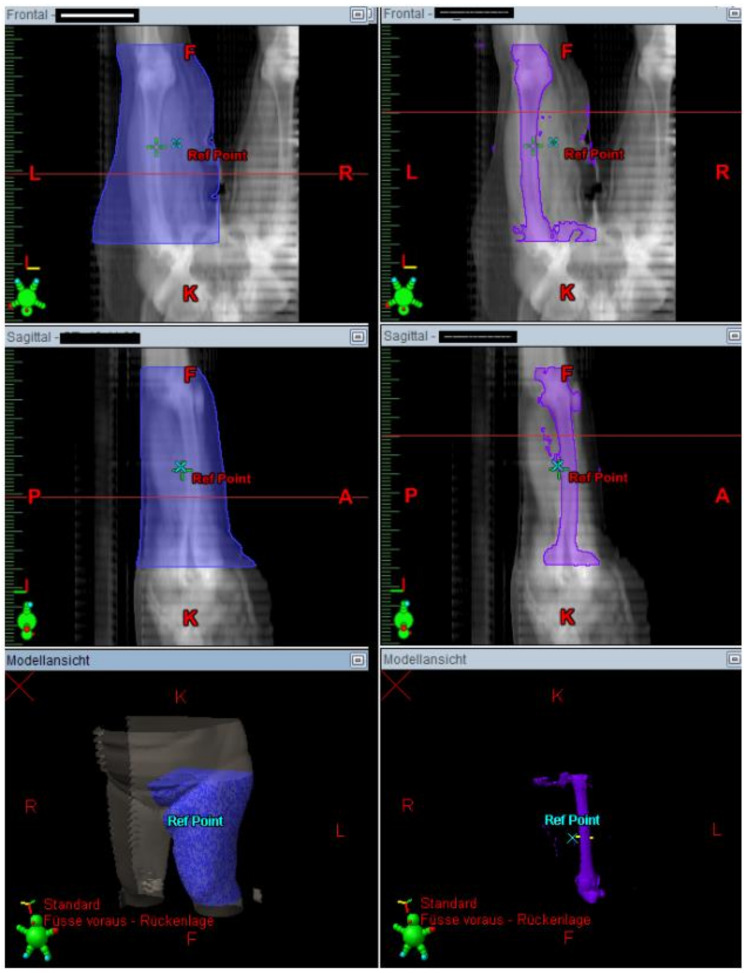
3D contouring of the extremity and bone to prepare bone subtraction from the extremity volume.

**Figure 4 cancers-13-02113-f004:**
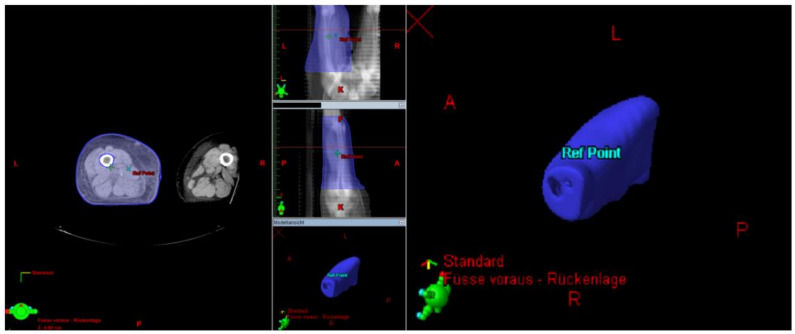
Subtracting the bone volume from the extremity volume.

**Figure 5 cancers-13-02113-f005:**
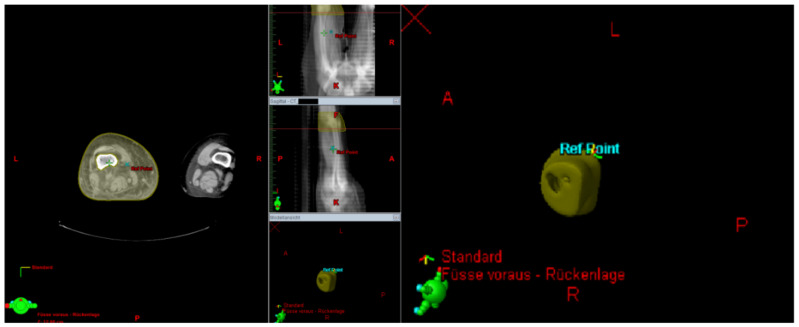
Contouring the area of the narrow lymph-sparing volume with 10 cm length.

**Figure 6 cancers-13-02113-f006:**
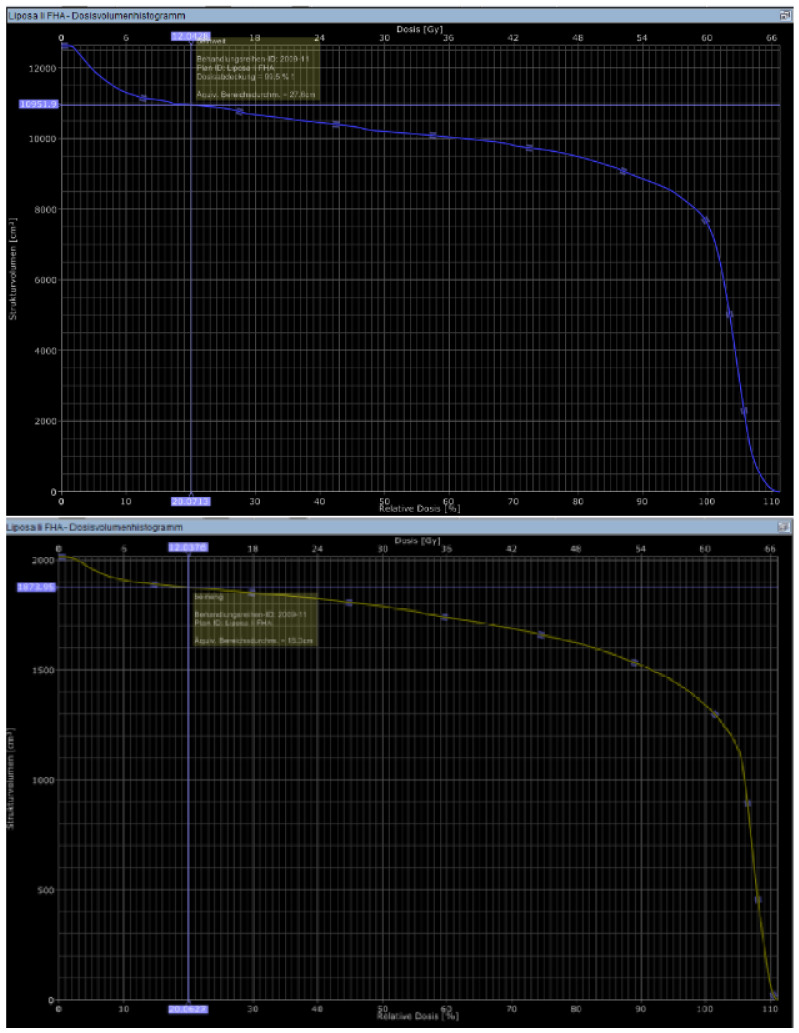
Calculating the volume receiving 20% of the dose in DVH (LSV) in the whole extremity (blue) and the narrowest part of the lymph-sparing volume (yellow).

**Table 1 cancers-13-02113-t001:** Patient and treatment characteristics.

Characteristic	Value	Percentage/Range
Gender:		
Male	16	47%
Female	18	53%
Age:		
Median	53 y	
Range	21–80 y	
Tumor location:		
Lower extremity	32	94%
Upper extremity	2	6%
Histology:		
Liposarcoma	9	26%
Spindle cell soft tissue sarcoma	5	15%
Synovial sarcoma	4	12%
Leiomyosarcoma	4	12%
Fibrosarcoma	1	3%
Myxofibrosacoma	4	12%
Alveolar sarcoma	1	3%
Soft tissue sarcoma (not otherwise specified)	1	3%
Pleomorphic sarcoma	5	15%
Histologic differentiation FNCLCC = Fédération Nationale des Centres de Lutte Contre Le Cancer [13]:		
Grade 1	6	18%
Grade 2	4	12%
Grade 3	24	70

Stage:		
Stage I	5	15%
Stage II	9	26%
Stage III	20	59%

Resection:		
Intralesional	1	3%
Marginal	15	44%
Wide	18	53%
Treatment parameters:		
Med. radiation dose (range), Gy	66.3 Gy (50.4–70.2 Gy)	
Med. fraction dose (range), Gy	1.8 Gy (1.8–2 Gy)	
Boost	27/34	79%
Med. follow up, months (range)	40 (3–223)	
Lymphedema ≥ Grade 2:		
Acute	5	15%
Chronic	17	50%
Acute or chronic	24	71%
Lymph-sparing quotient:	2YLEFS	*p*
≤0.2	0%	0.006
>0.2	39%	
Lymph-sparing quotient:	HR (95% CI)	*p*
/% increase	0.964 (0.93–1.0)	0.05

## Data Availability

The data presented in this study are available on request from the corresponding author. The data are not publicly available due to privacy.

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
