# Peer review of "The Lymph-Sparing Quotient: A Retrospective Risk Analysis on Extremity Radiation for Soft Tissue Sarcoma Treatment"

_cancers, 2021, doi:10.3390/cancers13092113_

Round 1

Reviewer 1 Report

The authors should be commended for adding to this important body of work, most applicable to providers caring for patients with (or a history of) extremity sarcomas.

A few minor recommendations:

In the methods section:

-consider defining why the 2-year time point was selected; move the description of the tiered lymphedema grading system from results to methods (and explain, e.g. are they using the Sterns-Rating scale or their own definition).

In table 1, the acronyms/initials should be spelled out in a footnote for the non-sarcoma reader (WTS, NOS). Did they inadvertently shorten pleomorphic sarcoma (pleomorph). Similarly the grading system (assume FNCLCC).

There should be consistency with the abbreviations. In the abstract 2 year lymphedema free survival is abbreviated as 2-year LEFS but in the rest of the paper including table 1, listed as 2YLEFS.

I am curious about the percentage of those with acute lymphedema who went on to develop chronic lymphedema. Is it a good predictor?

The authors appropriately highlight their small numbers and lack of testing of interobserver variability for estimating LSV. However they pointed out in their cohort there was no difference in lymphedema between upper and lower extremity. Suggest similarly pointing out the small numbers of upper extremity (2 of their 34) similarly impacted the ability to note the difference in lymphedema rate. In the discussion it would also be good to highlight the variability in grading and defining radiation induced lymphedema.

Author Response

Dear Reviewer,

thanks a lot for the rapid and detailed review on our article Submission.

To begin with, I will answer the comments and suggestions Point by Point and will conclude to what extent they led to a Change in the article itself.

1.) Why is the LEFS limited to 2 years in our study?

We also discussed this point before Submission The median follow up which is stated with 40 months represents a range of 3 months up to over 10 years so the decision was made to reduce the LEFS time in order to include as many patients as possible that reached a follow up of 24 months. We will add that in the Methods for Explanation.

“Since the 40 year median follow up represents a range from 3 to 223 months we decided to limit the endpoint to a 2 year period after therapy, to ensure most patients reached this follow up period." Page 2 methods line 8-11.

2)Placement of lymphedema Rating scales in the article.

We placed our procedure of grading lymphedema in the Methods sections. As a result of your Suggestion we also decided to extend and describe the differences of the lymphedema grading System in the different studies more spefically in the discussion.

3) Surely we will Change the use of abbreviations in the table and for 2YLEFS as suggested. (table 1 WTS; NOS, and grading FNCLCC)

4) Do Patient with acute lymphedema more commonly have chronic lymphedema?

Very interesting question and aspect. In the end the results did not show a significant correlation. From the 12 patients with acute lymphedema 6 went on to have a chronic lymphedema.

5)Surely we can underline that due to the small number of upper extremity sarcoma the results in comparison to lower extremity sarcoma cannot be statistically concluded. Also we will stress and discuss the different lymphedema grading scales with the difficulty of having different Treatment modalities. (page 10 in discussion)

Kind regards,

the authors

Reviewer 2 Report

It is interesting. The figures need to be more qualified and clearer.  

Author Response

Dear reviewer,

Thanks a lot for your suggestions.

We will make changes in the subtitles of figures and tables.

Regards,

The authors

Reviewer 3 Report

The authors describe the lymph-sparing-quotient – a retrospective risk analysis on extremity radiation for soft tissue sarcoma treatment. My comments to revise the result and discussion section.

3. Results. The authors mention "Acute lymphedema (within three months after radiation) was seen in 12 patients. Most of them were grade 1 lymphedema (7=58%), while five patients (42%) had a grade 2 or 3 acute lymphedema":
How many patients were affected directly after the operation? 

How many patients suffered from lymphocele or lymph fistula?

4. Discussion:

  • Please discuss the appearance of lymphocele or lymph fistulas
  • Please cite the possible intraoperative LYMPHA Technique to Prevent Secondary Lower Limb Lymphedema.
    Boccardo F, Valenzano M, Costantini S, Casabona F, Morotti M, Sala P, De Cian F, Molinari L, Spinaci S, Dessalvi S, Campisi CC, Villa G, Campisi C
  • Please discuss possible lymph surgery operation (post operation)
  • Please discuss manual lymphatic drainage after sarcoma operation and during Rtx
  • Please discuss Lymphangioarcoma as risk and cite: Lymphangiosarcoma: Is Stewart-Treves Syndrome a Preventable Condition? Felmerer G, Dowlatshahi AS, Stark GB, Földi E, Földi M, Ahls MG, Ströbel P, Aung T. Lymphat Res Biol. 2016 Mar;14(1):35-9. doi: 10.1089/lrb.2015.0006. Epub 2015 Nov 19

Author Response

Dear reviewer,

thanks a lot for the rapid and detailed review on our article Submission.

To begin with, I will answer the comments and suggestions Point by Point and will conclude to what extent they led to a Change in the article itself.

  1. Of the 34 selected patients no patient suffered from an acute postoperative lymphedema, lymphfistula or lymphocele. We will add this in the results. (page 7) However, we cannot deny the possibility of a postoperative reduction of lymph vessel capacity without a clinically visible lymphedema.
  2. Thanks a lot for the information on surgical techniques for preventing lymphedema. We will cite and discuss options of LYMPHA as a factor of reducing the therapy related risk of lymphedema We have also now added surgical options. (page 10)
  3. Conservative options for treating lymphedema especially decongestive therapy have been added. (page 10)
  4. Lymphangiosarcoma as a rare complication including the publication that was mentioned is now included. (page 10)

We hope to fulfil your expectations with our revision.

Thanks a lot and regards,

The authors